# The Role of Post-Operative Radiotherapy for Non-Spine Bone Metastases (NSBMs)

**DOI:** 10.3390/cancers15133315

**Published:** 2023-06-23

**Authors:** Sherif Ramadan, Andrew J. Arifin, Timothy K. Nguyen

**Affiliations:** Department of Radiation Oncology, London Health Sciences Centre, London, ON N6A 5W9, Canada; sherif.ramadan@lhsc.on.ca (S.R.); andrew.arifin@lhsc.on.ca (A.J.A.)

**Keywords:** stereotactic body radiotherapy, non-spine bone metastases, post-operative radiotherapy (PORT)

## Abstract

**Simple Summary:**

Many cancers can metastasize to bones and require surgical intervention to repair or prevent fractures. Radiation post-surgery has become a standard treatment to reduce the risk of local tumor recurrence. This article reviews the use of radiation after surgery for bone metastases outside of the spine. We discuss prior research and common practice patterns in this field. We found that post-operative radiation can reduce the risk that cancer will return to the same area and can improve patients’ pain and function. Radiotherapy treatments are commonly delivered in five to ten treatments and should ideally encompass the entire hardware inserted at the time of surgery. Studies suggest that timely radiation therapy can lead to improved outcomes for patients. The specific treatments utilized should be guided by both patient and disease factors. Further studies are needed to help guide the specific radiation techniques and doses that are used.

**Abstract:**

Non-spine bone metastases (NSBMs) can cause significant morbidity and deterioration in the quality of life of cancer patients. This paper reviews the role of post-operative radiotherapy (PORT) in the management of NSBMs and provides suggestions for clinical practice based on the best available evidence. We identified six retrospective studies and several reviews that examined PORT for NSBMs. These studies suggest that PORT reduces local recurrence rates and provides effective pain relief. Based on the literature, PORT was typically delivered as 20 Gy in 5 fractions or 30 Gy in 10 fractions within 5 weeks of surgery. Complete coverage of the surgical hardware is an important consideration when designing an appropriate radiation plan and leads to improved local control. Furthermore, the integration of PORT in a multidisciplinary team with input from radiation oncologists and orthopedic surgeons is beneficial. A multimodal approach including PORT should be considered for an NSBM that requires surgery. However, phase III studies are needed to answer many remaining questions and optimize the management of NSBMs.

## 1. Introduction

### Background

Bone metastases are a common part of many cancers’ natural history. With advances in systemic therapies, the number of patients who will present with bone metastases during their disease course is increasing [1,2]. These metastases can have heterogeneous presentations and can affect both the axial and appendicular skeleton [3]. For these patients, treatment can consist of surgery, systemic therapy, and/or radiotherapy [4]. However, there is significant nuance around the timing and delivery of these treatments. Among the factors that may affect a patient’s treatment course, the location of the bone metastases is particularly influential. For example, asymptomatic bone metastasis in a weight-bearing bone, such as the femur, may warrant radiotherapy to reduce the risk of fracture from continual tumor progression. Conversely, asymptomatic bone metastasis in the sternum could be reasonably observed and treated in the future if symptoms develop.

In the context of radiation oncology, the bone metastases literature has a predominant focus on spinal metastases. The disproportionate attention toward spinal metastases may be partly due to the technical and safety concerns involved with delivering stereotactic body radiotherapy (SBRT) adjacent to the spinal cord. As the spine SBRT literature grew steadily, studies examining osseous metastases in the remainder of the skeleton lagged behind, particularly with respect to modern radiotherapy techniques, such as SBRT. These other osseous metastases have become known simply as non-spine bone metastases (NSBMs), which may be an oversimplification given the heterogeneous nature of the lesions. The most common areas of NSBMs are the pelvic bone, long bones (humerus/femur), and ribs [5,6]. Currently, there are limited studies informing the best treatment pathway for these patients. The vast majority of NSBMs are currently managed with analgesia, radiotherapy, or systemic treatments, but surgical fixation is indicated when there is instability of the bone with a high risk of fracture or an acute fracture [7].

Palliative radiotherapy alone has a well-established role in pain and symptom management of bone metastases [8,9,10,11,12]. The recent ESTRO ACROP guidelines recommend the use of palliative radiotherapy for uncomplicated bone metastases [13,14]. A taskforce study by Lopez-Campos et al. gave some guidance regarding this decision point [15]. Factors they considered in their decision-making process included CT-based structural rigidity assessment of the bone, percent of body weight involved in limb bearing, circumferential involvement of cortical bone on CT, and the Harrington and Mirels’ scores [16]. The commonly used Mirels’ score has been previously utilized to help guide when prophylactic fixation would be beneficial for bone metastases with high sensitivity and specificity [17]. As seen in Table 1, practitioners assess the location of the lesion, size, type, and degree of patient pain to calculate a cumulative score. This score estimates fracture risk at six months after radiation and can identify those who may benefit from prophylactic fixation. The taskforce study found that if there is a high Mirels’ score, significant concerns around structural rigidity, or a large burden of disease, it is generally recommended to consider upfront prophylactic surgery prior to radiotherapy.

In patients undergoing upfront surgery, the orthopedic surgeon will decide on the type of surgical procedure. The goals of surgery are to reduce pain, improve functionality, and prevent the risk of impending fracture. The specifics of the procedure are decided based on the anatomical location, patient life expectancy, number, and type of bone metastasis. If there is only one lesion, extensive resection, reconstruction, and modular prosthesis is generally preferred. For multiple metastases, the algorithm is more complicated and dependent on the bone involved and whether a fracture is present. This can lead to either resection and reconstruction with prostheses, resection arthrodesis, or preventative osteosynthesis. When these operations are performed, a variety of hardware can be utilized, including prostheses, intramedullary nail fixation, plates, and polymethyl methacrylate [18]. The specific hardware utilized has various implications for subsequent radiation planning.

In the post-operative setting for spine metastases, it has been well-validated that PORT leads to improved pain control and a reduced risk of local recurrence; it is associated with a low toxicity profile and improved patient outcomes [19,20,21,22,23]. However, there is a paucity of literature pertaining to PORT for NSBMs. The goal of this study is to review the indications, evidence, outcomes, and technical details of delivering PORT for NSBMs. This study will also discuss when to refer for PORT and when to consider re-irradiation, as well as important patient outcomes.

## 2. Evidence for Post-Operative Radiotherapy in Non-Spine Bone Metastases

As seen in Table 2, there are six retrospective studies on PORT for NSBMs [24,25,26,27,28,29]. To our knowledge, there are no prospective studies completed to date. The findings of these studies can be seen in Table 3 and Table 4 [15,30,31,32,33]. Additionally, there was a systematic review performed on the topic as well as a survey of radiation oncologists in Japan [34,35].

### 2.1. Surgical Techniques and Fracture Type

The literature on NSBM is quite uniform from a surgical perspective. Most of the studies discussed fractures of the femur. However, there was still a fair number of humerus fractures included, with a small number of knee/tibia fractures. These fractures received intramedullary nails, with plates comprising a smaller percentage. Moreover, most patients had pathological fractures, although some had impending fractures.

Overall, the type of surgery, location of the fracture, or type of fracture were not heavily analyzed in these retrospective studies. Epstein et al. and Rosen et al. both found that there was a correlation between the use of intramedullary hardware failure and whether a patient received PORT that covered the entire hardware [26,29]. This supports the hypothesis that the insertion of hardware can cause micro seeding through the length of the prosthetic, leading to local failures if not properly irradiated.

Additionally, very few patients in these studies required repeat surgical fixation after PORT due to the durable local control. This is achieved through radiation stabilizing the metastatic lesion and helping to maintain the alignment of the surgical prosthesis.

### 2.2. Indications for Post-Operative Radiation Therapy—Local Control

The current standard of care in the NSBM consists of PORT and is based on several historical studies, which demonstrated improved local control with the addition of radiotherapy.

Townsend et al. were the first to demonstrate a benefit for PORT in NSBMs [27]. The authors primarily examined femoral metastases but they also included some humeral metastases. They found that between 30 and 45 Gy of radiation delivered 14 days post-operatively led to an increased overall survival and local control in patients. The survival difference was 12.4 vs. 3.3 months and was theorized to be the result of a greater proportion of the disease growing slower in the radiation arm. They also noted that 15% of patients who received surgery versus only 3% of patients who had surgery plus PORT required a second surgery, demonstrating the benefit of PORT. Similar results were seen in the remaining studies identified [25,28,29], except for one study that found a non-significant trend toward higher rates of local failure following PORT [24]. These studies suggest that PORT leads to improved local control; they are the basis for the routine clinical use of this strategy.

### 2.3. External Beam Radiation Dose and Technique

#### 2.3.1. Dose

All six of the studies reported different dose fractionation schedules. Commonly used regimens included 8 Gy in 1 fraction, 20 Gy in 5 fractions, and 30–45 Gy in 10 fractions. A survey of radiation oncologists in Japan [34] found that over 50% of respondents preferred to use 30 Gy in 10 fractions post-operatively. It was felt that this longer course of radiation would lead to improved local control and prevent hardware failure compared to 20 Gy in 5 fractions or shorter regimens. Rosen et al. also demonstrated that higher doses led to improved local control [29]. Following surgical decompression for spinal metastases, there are randomized data showing a benefit of adjuvant radiotherapy to a dose of 30 Gy in 10 fractions [12]. As such, The ESTRO-ACROP guidelines have recommended 30 Gy in 10 fractions as the standard dose schedule for spinal PORT [13]. While similar high-level evidence does not exist for the NSBM PORT, the evidence for spine PORT can be reasonably extrapolated. Similar to the spine setting, there is typically residual untreated disease following surgical intervention for NSBM, as surgery will often focus on stabilization rather than comprehensively resecting or ablating gross disease. Therefore, it is worth considering 30 Gy in 10 fractions for the PORT NSBM as a starting point. Shortened dose schedules may be more appropriate for certain patients such as those with poor performance status, travel limitations, and/or who need urgent radiotherapy (e.g., minimize delays to starting systemic therapy).

#### 2.3.2. Radiotherapy Target Volumes

The target volumes when irradiating NSBMs can vary depending on the anatomical region. However, some general principles apply broadly in the treatment of these metastases when using conventional external beam radiotherapy. Firstly, we use two beams in a parallel and opposed manner to ensure a uniform dose is delivered at the depth of the tumor. This two-beam arrangement can be seen in Figure 1A. The next step is to ensure that an adequate volume of bone and tumor is covered. The goal is to cover the entire resection cavity and extend the radiation field to include the entire surgical hardware as well as any visibly affected bone. Additionally, an increased margin around the areas of interest of 1.5–2 cm is added. This accounts for the rapid fall off of radiation dose at the edge of the beam, also known as the penumbra, as well as the variation in patient position and motion known as the planning target volume (PTV). Tissue within the field that is not desired to be irradiated is blocked off from receiving the dose with the use of multi-leaf collimators, which are seen as the white lines in Figure 1B. Typically, this would include soft tissue or uninvolved bones/organs.

The studies reported the proportion of cases having full coverage of hardware from 39% to 97%. Specific clinical factors and user preferences are likely to drive this significant variation in hardware coverage. The series with 97% of cases having full coverage of the hardware had a low incidence of second surgery or re-irradiation [28].

Rosen et al. conducted a large retrospective analysis involving 145 lesions that received post-operative radiation. The study found that patients receiving radiation to the entire surgical hardware experienced fewer local recurrences, 12% versus 21% at one year, and 16% versus 41% at two years [29]. Their analysis utilized propensity score matching and found a statistical association between whole hardware coverage and reduced risk of recurrence. Epstein-Peterson et al. analyzed data from 82 bone metastases that underwent post-operative radiotherapy. Their study involved, on average, 71% of the surgical hardware being covered in the radiotherapy field. They then had fourteen cases with local progression. Moreover, in the multivariable analysis, they found that increased coverage of the surgical hardware by the RT field led to a reduced risk of local failure, which had to be salvaged with a second surgery [26].

Ultimately, both Rosen et al. and Epstein-Peterson et al. showed that ensuring surgical hardware was included within radiotherapy volumes led to a statistically significant improved reduction in local recurrence. The association of hardware coverage and reduced local recurrence is thought to be related to tumor seeding along the operative hardware when the components are inserted and fixated [36]. Despite these findings, the ESTRO guidelines do not make a definitive recommendation regarding treatment volumes and inclusion of the surgical hardware in the post-operative setting [13]. However, retrospective evidence available to date would suggest that when technically feasible, the entirety of the surgical hardware and implants should be included in the target volumes for PORT in NSBMs.

### 2.4. Timing

The timing of PORT also varies among studies. The median time to treatment ranged from 14–41 days, though several studies did not report these data. This does show a significant difference between when patients are treated surgically to the time of their radiotherapy. Epstein et al. performed a multivariable analysis on the 52 lesions treated in their study. They found an increased risk of local recurrence with increasing time between surgery and PORT [26]. Their analysis had a significant *p*-value of 0.01 and a confidence interval of 1.01–1.06.

When determining the time to treat patients, there is a fine balance between prompt delivery of radiotherapy and allowing for adequate surgical wound healing. The time to PORT ultimately must be made on a case-by-case basis, and factors to consider include disease histology, degree of residual disease, wound healing, and the type of procedure that was performed. Patients receiving minimally invasive fixation will be able to receive radiation earlier than those who had a large open operation. Depending on the nature of the surgery, at least one to two weeks should be allowed for wound healing, and treatment starting within five weeks from surgery is recommended [25,26,27,29]. A comprehensive multidisciplinary approach between orthopedic surgeons and radiation oncologists should occur to help facilitate the timely delivery of this multimodality treatment. If possible, the incision should be assessed by the surgeon prior to starting radiotherapy.

### 2.5. Stereotactic Body Radiotherapy

Evidence for SBRT in the post-operative setting is emerging. There was only one study in our review that included patients treated with SBRT [29]. Twelve lesions received SBRT, although the authors noted there was no institutional policy to guide the treatments, and all twelve patients received significantly different approaches and treatment volumes. Aside from this report, there is an absence of literature examining SBRT for PORT for NSBM. However, we can use data from analogous settings to provide guidance for this treatment approach.

In the non-operative NSBM setting, 30–35 Gy in 5 fractions has been recommended by some experts [15,30,31]. This regimen has been found to be effective in providing local control and symptom relief. A study by Madani et al. [37] demonstrated high rates of local control with limited fracture risk when using SBRT doses of 30–50 Gy in 5 fractions for long bones. One can likely also use these dose schemes in the post-operative setting, but obtaining guidance around treatment volumes from these studies is more difficult.

One challenge with post-operative SBRT in a long bone is the need to cover the entire surgical hardware. It can be prohibitive to treat such a large volume to a full SBRT dose. A reasonable approach would be to use a simultaneous integrated boost technique where the surgical cavity, residual disease, and entire surgical hardware receive a moderate subclinical dose (e.g., 20–25 Gy in 5 fractions), and the residual disease +/− the surgical cavity is boosted to a higher dose (e.g., 30–35 Gy in 5 fractions). This fractionated approach can also help to avoid the moderate fracture risk seen in single-fraction SBRT in the spine.

An example post-operative SBRT plan in the re-irraditation setting can be seen in Figure 1C.

### 2.6. Functional Status and Pain

When utilizing radiation in the pre-operative or non-operative setting, it has been well-documented that radiation is an effective means of reducing bone metastases pain [38,39,40]. There are various fractionation schedules that have been utilized, including 8 Gy in 1 fraction, 20 Gy in 5 fractions, and 30 Gy in 10 fractions being the most used. However, there has been no significant difference in the level of pain control that patients achieve between these doses. The only major difference is that review studies have found that patients receiving single-fraction treatment have a higher incidence of requiring re-irradiation [11]. The efficacy of radiation in achieving pain control has also been found to hold true with the utilization of SBRT [41].

Several studies reported the impact of radiation on patient function and pain in the PORT setting [24,27]. Van Geffen et al. showed that 79% of patients had improved mobility, and 60% did not require any analgesia. However, they had a small cohort of patients who received PORT. More recently, Admietz et al. and Drost et al. found that almost 93% of patients had a normal functional status after PORT [25,28].

### 2.7. Toxicity

Overall, the conventional radiotherapy doses previously discussed are associated with minimal toxicities. This has been borne out in both palliative and post-operative settings. Epstein-Peterson et al. and Van Geffen et al. found that fewer than 20% of patients had local acute side effects, many of which were related to their surgical procedure [24,26]. These side effects included wound infection and bleeding. They noted there was not any difference in toxicity among the anatomical regions that were treated. In general, there is minimal toxicity from radiotherapy of the extremities. If there is a bowel or bladder in the treatment, field patients can experience transient diarrhea and urinary frequency/dysuria. The most common side effect from this type of radiation therapy is the risk of pain flare, where patients get an acute and transient worsening of their pain before starting to experience relief 1–2 weeks post-treatment. In historical non-operative studies, approximately 16% of patients have been seen to experience pain flare at 48 hours post-treatment [42].

### 2.8. Re-Irradiation

Re-irradiation is a difficult situation for radiation oncologists given concerns about increased toxicity. In our review, only Drost et al. examined re-irradiation, noting that 9.5% of patients required a second treatment for repeat or no relief of pain [28].

A systematic review of re-irradiation showed that it provides improved local control and is associated with a 68% chance of pain relief [43]. The doses used for retreatments vary, but 8 Gy in 1 fraction is commonly used and is recommended in the ESTRO guideline [13]. SBRT has gained popularity in the re-irradiation setting, particularly for radioresistant tumors, as a way to give more precise and ablative doses to areas of recurring disease or symptoms [44,45]. When SBRT is used, typical doses range from 30 to 35 Gy in 5 fractions.

### 2.9. Clinical Example

A clinical example has been included in Figure 1 as a demonstration of management principles. This patient had a pathological fracture of the right femur, which required reconstruction. The patient then received PORT covering the entire surgical hardware to a dose of 20 Gy in 5 fractions. Unfortunately, this patient fell within the minority and had local recurrence despite this PORT. He subsequently had SBRT to 25 Gy in 5 fractions with a boost to 35 Gy at the site of gross disease.

## 3. Future Areas of Interest

There are currently two prospective randomized trials looking at PORT for NSBMs in the clinical trials database (clinicaltrials.gov identifier NCT02705183 and clinicaltrials.gov identifier NCT04109937) [46,47]. One is examining PORT for long bone metastases and has an unknown status according to clinicaltrials.gov. The trial was estimated to finish in 2019. The second trial, known as EXPLORE, has not yet started recruiting and has an estimated completion date of 2026. This phase III study aims to compare surgery alone versus surgery and PORT (20 Gy in 5 fractions or 30 Gy in 10 fractions). The outcomes of this study will include the requirement of a second surgery, reirradiation, performance status, post-operative pain and analgesic use, local progression, quality of life, functional status, overall survival, and cost-effectiveness. This is a promising study that could help to address many of the questions raised in this review.

Future work is needed to expand the literature in this area. A definitive study investigating the timing, dose, and extent of radiation field coverage would help guide clinicians to optimize treatment. Additionally, prospective trials examining the safe and optimal SBRT dose in the post-operative setting would be beneficial in guiding the utilization of more precise and higher doses of radiation. Moreover, a detailed investigation on the utilization of re-irradiation could help give patients an additional safe line of therapy moving forward. Although this review focused on PORT, there would be value in determining whether preoperative radiation provides equal benefits for patients.

## 4. Conclusions

The present article reviews the role of PORT for NSBMs. There are six retrospective studies that have investigated the topic and no prospective or randomized control trials. There is also a single systematic review that had only identified two papers at the time of publication. Across the six studies, PORT appears to provide a benefit for patients in terms of local control, pain relief, and improved functional status, and has little associated toxicity. In terms of radiation technique, treatments have historically been delivered within 5 weeks but can be delivered once adequate surgical healing has occurred. There are also trends to show that more durable local control is achieved when the radiotherapy plan includes the entire surgical hardware. Currently, external beam radiotherapy is delivered in a variety of fractionation schedules, including 20 Gy in 5 fractions and 30 Gy in 10 fractions. Moving forward, we expect to see an increased role of SBRT for NSBMs in the post-operative setting. Further research in this area is required to better guide clinical decision-making and treatment planning.

## Figures and Tables

**Figure 1 cancers-15-03315-f001:**
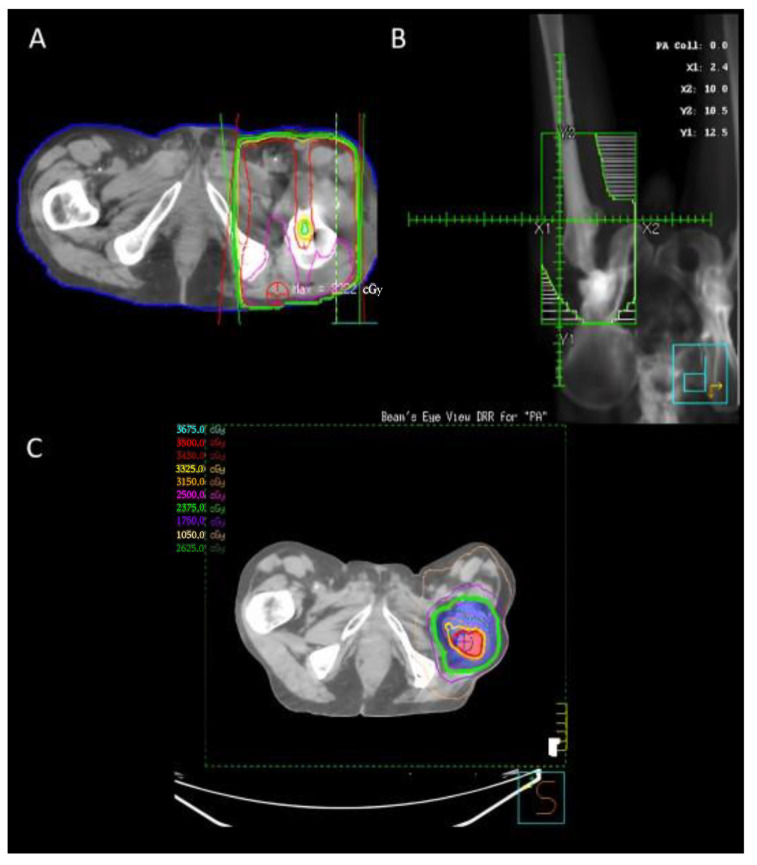
Radiation plans for a patient who underwent two treatments. (**A**,**B**) A 20 Gy in 5 fraction parallel opposed pair post-operative plan for a patient who underwent a right femur reconstruction for bone metastases. The radiation field includes the full hardware. (**C**) The patient had a recurrence within a year and was re-irradiated to 25 Gy in 5 fractions and a simultaneous in-field boost to 35 Gy.

**Table 1 cancers-15-03315-t001:** Mirels’ score for estimating extremity fracture risk and management [17].

	Score		
	1	2	3
Location	Upper extremity	Lower extremity	Periotrochanteric region
Pain	Mild	Moderate	Severe
Lesion	Blastic	Mixed	Lytic
Size	<1/3 of bone width	1/3 to 2/3 of bone width	>2/3 of bone width
		Fracture risk	Recommendation
Total Score	≤7	0–4%	Safe to irradiate with minimal risk of fracture
	8	15%	Consider prophylactic fixation
	≥9	>33%	Prophylactic fixation indicated

**Table 2 cancers-15-03315-t002:** Summary of studies of post-operative radiotherapy for non-spine bone metastases.

Study	Rosen et al., 2021 [29]	Adamietz Wolanczyk, 2018 [25]	Drost et al., 2017 [28]	Epstein Peterson et al., 2015 [26]	Van Geffen et al., 1997 ** [24]	Townsend et al., 1995 [27]
Study Design	Retrospective	Retrospective	Retrospective	Retrospective	Retrospective	Retrospective
Number of Lesions	*n* = 145	*n* = 68	*n* = 74	*n* = 52	*n* = 27	*n* = 35
Location	Long Bones	- FE (81%) - HU (19%)	- FE (79.7%) - HU (13.5%) - Knee (6.8%)	- LE (46%) - UE (17%) - Spine (37%)	- FE (59%) - HU (20%) - Spine (8%) - Pelvis (7%)	- FE (91%) - HU (6%)
Operative Technique	IM (77%) Plate (23%)	IM (55.8%) Plate (44.1%)	Surgical Fixation	ORIF (13%) IM (45%) Spine (37%)	IM (23%) Plate (51%) Other (20%)	NR
Radiation Dose	EBRT (92%) (20–30 Gy) SBRT (8%)	EBRT (Mean) (34 Gy/7.8)	EBRT (30/10, 20/5, 8/1, other)	EBRT (30/10, 20/5, 24/6, 8/1)	NR	EBRT (30–45 Gy)
Fracture Type	- PF and IF	- PF (70.5%) - IF (30%)	- PF and IF	- PF (54%) - IF (24%) - No # (17%)	- IF (100%)	- PF (51%) - IF (49%)
Local Control	70%	95.6%	83%	83%	20%	91.2%
Hardware Coverage	52%	52.9%	97.3%	26%	NR	39%
OS (mean months)	NR	16.3	NR	6.7	15	PORT 12 None 3.3
Pain /Function	NR	Normal Function (93%)	General Improvement	NR	Improved Function (79%) Pain-free (60%)	Normal Function (51%)
Toxicity	NR	NR	NR	Local (17%)	Local (14%)	NR
Follow-up (median months)	29.5	16.3	NR	11.5	NR	10.7
Time from operation to radiation (mean days)	41	33.6	NR	20	NR	14

NR = not reported, NSR = did not separately report patients who received PORT, PF = pathological fracture, IF = impending pathological fracture, UE = upper extremity, LE = lower extremity, HU = humerus, FE = femur, IM = intramedullary.

**Table 3 cancers-15-03315-t003:** Key findings from post-operative bone metastases studies.

Study	Key Findings
Rosen et al., 2021 [29]	Coverage of the entire radiation hardware was associated with reduced local recurrence.
Adamietz and Wolanczyk, 2018 [25]	Pre-operative status was a strong prognostic factor for improved recovery. Rapidly growing tumor pathologies were found to have worse recovery outcomes.
Drost et al., 2017 [28]	Re-demonstrated the benefit of PORT. Few incidents of second surgery or re-irradiation.
Epstein-Peterson et al., 2015 [26]	Full coverage of operative hardware and reduced time to initiate PORT led to a reduction in local recurrences.
Van Geffen et al., 1997 [24]	The only negative study of PORT. PORT impaired local recurrence and implant failure, but this was not statistically significant.
Townsend et al., 1995 [27]	Demonstrated an improved functional status in patients who received PORT.

**Table 4 cancers-15-03315-t004:** Periphery studies of interest, guidelines, and SBRT usage.

Study	Study Type	Radiation Type	Summary
Lopez et al., 2022 [15]	Guideline (Not PORT)	SBRT	SBRT for non-spine bone metastases. Recommendations for lower and upper extremities on radiation vs. surgery. Considered imaging factors, % of bone involvement, Mirels’ score, and size.
Nguyen et al., 2022 [30]	Guideline	SBRT	Practice guidelines for SBRTs in long bones. A value of 35 Gy/5 was the most common fractionation regimen used. Contouring recommendations for CTV.
Ito et al., 2021 [31]	Prospective (Not PORT)	SBRT	Investigated SBRT (30 or 35/5) for femoral humeral and radial bones. Effective at pain palliation.
Pinta et al., 2020 [32]	Literature Review	SBRT	Detailed overview of SBRT in non-spine bone metastases. Doses have been found to vary from 24–50 Gy in 3–5 fractions. SBRT was found to result in the local control of approximately 90%. Limited direct comparison to EBRT.
Kubota et al., 2020 [34]	Survey	EBRT	Practice patterns in Japan. A total of 50% of the respondents preferred 30 Gy in 10 to achieve improved local control and reduce incidence re-irradiation. The entire orthopedic prosthesis was included in 74% of the respondents.
Elhammali et al., 2019 [33]	Retrospective	EBRT	Investigated EBRT post-operatively for multiple myeloma. Full coverage of the operative hardware provided improved control.

## Data Availability

The data is contained within this article.

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
