# Peer review of "The Role of Post-Operative Radiotherapy for Non-Spine Bone Metastases (NSBMs)"

_cancers, 2023, doi:10.3390/cancers15133315_

Round 1

Reviewer 1 Report

# The role of postoperative radiotherapy for non-spinal bone metastases (NSBM): Review and Recommendations

## Summary
This article presents the results of a review of the available evidence on the role of postoperative radiotherapy (RT) following surgery for non-spinal bone metastases (NSBM).
I respectfully suggest that the paper should not be accepted for publication in its current form and recommend extensive major revisions.

## General comments
My impression of this manuscript is moderate. The main problems are:

I recommend that the authors rewrite the manuscript and focus only on the systematic review. The important guidelines on the role of RT in bone metastases were published last year and should also be discussed (https://doi.org/10.1016/j.radonc.2022.06.002 and https://doi.org/10.1016/j.radonc.2022.05.024). The issue of postoperative RT in NSBM was mentioned there and expert consensus recommendations were given based on the best available evidence.
I am not in favour of making recommendations and using very strong words (like "crucial") based only on retrospective studies (weak evidence). Frankly, I don't see a basis in this manuscript to make a recommendation.
In the case of a review, I recommend using the PRISMA guidelines (http://prisma-statement.org/?AspxAutoDetectCookieSupport=1) and showing the results (diagram + checklist).
Be more concise when rewriting, avoid unnecessary parts of the text. For example, you focus on NSBM; however, you wrote "A recent review by Faruqi et al found that postoperative SBRT in the spinal setting...". - I think it's very inconsistent and makes the whole text chaotic.
Figure 2 looks a bit strange, like a diagram from MS Word 2000. I think it's unnecessary.
All the tables are chaotic - please redesign them and make them clearer + improve the presentation considerably.

Reviewer 2 Report

Dear Authors,

The postoperative RT for none-spine bone metastasis is a relevant and challenging topic, as we do not have much data, as we have for spinal metastasis. Your work sheds more light into the field and every attemp to guide the clinicians to do better for such patients should be appreciated. 

I have some points that I wanted to share with you:

1. Although your focus is on postoperative setting, but it is worth to mention the publication from 2022 regarding SBRT for long bone metastasis (Guckenberger et al., Red Journal 2022).

2. It would be beneficial for the paper to make a more clear statement about including the surgical implants into RT field. It is daily clinical practice and from time to time there is unclearness about the extent of implants to be included in RT fields.

3. It is also worth to mention the work from Redmond et al, regarding postoperative spine SBRT from 2020 (phase II study).

Again thank you for this nice work. 

Round 2

Reviewer 1 Report

Thank you for implementing my comments. The manuscript has improved significantly. In my opinion, now it's suitable for publication.